# Del1 Is a Growth Factor for Skeletal Progenitor Cells in the Fracture Callus

**DOI:** 10.3390/biom13081214

**Published:** 2023-08-03

**Authors:** Yuxi Sun, Tatiana Boyko, Owen Marecic, Danielle Struck, Randall K. Mann, Tom W. Andrew, Michael Lopez, Xinming Tong, Stuart B. Goodman, Fan Yang, Michael T. Longaker, Charles K. F. Chan, George P. Yang

**Affiliations:** 1Department of Surgery, University of Alabama at Birmingham, Birmingham, AL 35233, USA; 2Division of Plastic and Reconstructive Surgery, Department of Surgery, Stanford University School of Medicine, Stanford, CA 94305, USArmann@stanford.edu (R.K.M.); tandrew@stanford.edu (T.W.A.); longaker@stanford.edu (M.T.L.); 3Department of Orthopedic Surgery, Stanford University, Stanford, CA 94305, USA; xingming@stanford.edu (X.T.); goodbone@stanford.edu (S.B.G.); fanyang@stanford.edu (F.Y.); 4Department of Bioengineering, Stanford University, Stanford, CA 94305, USA; 5Institute for Stem Cell Biology and Regenerative Medicine, Stanford University School of Medicine, Stanford, CA 94305, USA; 6Birmingham VA Medical Center, Birmingham, AL 35233, USA

**Keywords:** fracture healing, bone regeneration, skeletal progenitor cells, Del1, proliferation

## Abstract

Failure to properly form bone or integrate surgical implants can lead to morbidity and additional surgical interventions in a significant proportion of orthopedic surgeries. While the role of skeletal stem cells (SSCs) in bone formation and repair is well-established, very little is known about the factors that regulate the downstream Bone, Cartilage, Stromal, Progenitors (BCSPs). BCSPs, as transit amplifying progenitor cells, undergo multiple mitotic divisions to expand the pool of lineage committed progenitors allowing stem cells to preserve their self-renewal and stemness. Del1 is a protein widely expressed in the skeletal system, but its deletion led to minimal phenotype changes in the uninjured mouse. In this paper, we demonstrate that Del1 is a key regulator of BCSP expansion following injury. In Del1 knockout mice, there is a significant reduction in the number of BCSPs which leads to a smaller callus and decreased bone formation compared with wildtype (WT) littermates. Del1 serves to promote BCSP proliferation and prevent apoptosis in vivo and in vitro. Moreover, exogenous Del1 promotes proliferation of aged human BCSPs. Our results highlight the potential of Del1 as a therapeutic target for improving bone formation and implant success. Del1 injections may improve the success of orthopedic surgeries and fracture healing by enhancing the proliferation and survival of BCSPs, which are crucial for generating new bone tissue during the process of bone formation and repair.

## 1. Introduction

The human body’s ability to repair bone damage, as seen after fractures, is a crucial physiological response that ensures survival. However, 3–5% of the 7.3 million orthopedic surgeries that present at emergency rooms in the USA each year fail to heal properly, resulting in significant morbidity [1,2]. Proper bone formation is also critical for successful outcomes in various medical procedures, including spinal fusion, dental implant, and total joint replacement. Nearly 2% of total knee and hip arthroplasties fail within 10 years due to implant loosening, as reported by joint replacement registries [3]. 

Recent advancements have shed light on the pivotal role played by Skeletal Stem Cells (SSCs) and their downstream Bone, Cartilage, Stromal, Progenitor cells (BCSPs) in the process of chondrogenesis and osteogenesis during endochondral and intramembranous ossification in various components of the skeletal system [4,5,6]. BCSPs serve as an invaluable reservoir of multi-potent transit amplifying cells, which undergo a limited number of mitotic divisions and give rise to more lineage-restricted downstream progenitor cells, thereby expanding the pool of essential skeletal progenitor cells while simultaneously preserving the population of stem cells. This well-coordinated and efficient stepwise mechanism of proliferation highlights the critical role of BCSPs in the regenerative process of skeletal tissues [4,7]. Despite the critical role that BCSPs appear to play in skeletal tissue regeneration, their specific functions have only recently come under scientific scrutiny, and their precise mechanism of action remains largely unknown.

Del1 (Developmental Endothelial locus-1), a matricellular protein, exhibits high expression in various tissues, including vascular endothelial cells [8], hematopoietic cells [9], skeletal tissues [10], neural tissue [11], and adipose tissue [12]. In various tissues including hematopoietic cells and immune cells, Del1 has been identified as a driver of the expansion of progenitor cells [13,14]. In the context of hematopoietic progenitor cells, Del1 has been shown to promote their expansion and self-renewal, implying its involvement in maintaining and replenishing the hematopoietic cell pool. Similarly, Del1 has also been implicated in driving the expansion of the forebrain, indicating its potential role in neurogenesis and brain plasticity. Considering Del1 is highly expressed within the skeletal system, we examined its possible influence on skeletal progenitor cells (BCSPs) for the growth, maintenance, and repair of skeletal tissues such as bone and cartilage. 

Regarding skeletal tissue, our previous studies have shed light on the significance of Del1 in skeletal biology. Notably, the absence of Del1 in knockout mice resulted in exacerbated osteoarthritis in a model of traumatic osteoarthritis [15]. This finding strongly suggests that Del1 serves as a potent anti-apoptotic factor for chondrocytes, providing a protective function in maintaining cartilage integrity. Furthermore, investigations into periodontitis models have demonstrated the critical role of Del1 in bone maintenance [16,17]. Thus, while its role in regulating skeletal stem and progenitor cells is still being investigated, the existing evidence suggests its potential involvement.

We hypothesized Del1 is important in the response to injury and examined fracture healing in the KO mice as a model of skeletal injury. We find that the KO mice healed fractures with significantly less bone and smaller calluses compared to the wild-type (WT) mice. We show this was due to reduced expansion of BCSPs, but not SSCs with upregulated apoptosis. BCSPs from KO mice demonstrated decreased proliferation and formed fewer colonies in culture. The addition of exogenous Del1 restored normal proliferation rates in culture and led to more bone formation and larger calluses. In addition, human recombinant Del1 could promote the proliferation of aged human BCSPs. In summary, we showed that Del1 functions promote proliferation and prevent apoptosis of BCSP. 

## 2. Materials and Methods

### 2.1. Animal Care

Del1–LacZ knockin mice were previously described [8,15,18,19]. To characterize the LacZ expression, healing tibia was collected 4 weeks post-fracture. Specimens were fixed in 4% paraformaldehyde and placed in X-gal solution (400 μg/mL X-gal reagent (Invitrogen, Carlsbad, CA, USA)), 5 mM potassium ferricyanide, and 2 mM MgCl_2_ in 1x phosphate-buffered saline. Specimens were incubated at 37 °C in an incubator for 1 to 8 h until staining was apparent in test specimens but not control specimens, and post-fixed in 4% paraformaldehyde followed by embedding in paraffin, sectioning, and counterstaining with eosin. Wild type (WT) C57BL/6 mice were purchased from Charles River Laboratories (Wilmington, MA, USA). All animal procedures were approved by the Institutional Animal Care and Use Committees (IACUCs) of Stanford University and the University of Alabama at Birmingham. Animals were maintained at the Stanford University Comparative Medicine Pavilion and Research Animal Facility or the UAB Animal Resources Program in accordance with Animal Care and Use Committee determined guidelines (Stanford protocol #15966, UAB protocol #21355). 

### 2.2. Fracture Models

For tibial fractures, 6-week-old mice were anesthetized with aerosolized isoflurane. Buprenorphine was administered for anesthesia and the surgical site was prepared before the skin incision over the left anterior tibia. A transverse, mid-diaphyseal tibial fracture was made by a bone cutter. The skin was closed using a 6–0 nylon suture. 

For femoral fractures, 8-week-old mice were anesthetized with aerosolized isoflurane and were administered with buprenorphine for anesthesia. The surgical site over the femur was prepared before an incision was made from the groin extending to the knee. The patella was dislocated laterally to expose the distal femoral epiphysis. A 27.5 gauge needle was inserted into the medullary cavity with the needle rod remaining in situ for fracture healing guidance. A transverse, mid-diaphyseal fracture was made by micro-scissors. The patella was relocated, muscle was reapproximated, and the skin was closed using a 6–0 nylon suture. A minimum of three or more mice were used as the sample size of this study. 

### 2.3. Isolation of Skeletal Progenitors

The protocol was described previously [20]. Fractured and uninjured contralateral femur were harvested at 7 days post femoral fracture. The tissue was independently crushed with a mortar and pestle before enzymatic digestion with collagenase (2.2 mg/mL collagenase, 300 units/mL DNase, 0.1% bovine serum albumin, 0.1% Pluronic F-68, 2% 1 M HEPES, 0.11 mg/mL 2.5 M CaCl_2_, in M199). Each sample underwent three serial digestions at 37 °C for 25 min under gentle agitation. Dissociated cells were filtered by 40 μm nylon mesh and washed with FACS Buffer (2% fetal bovine serum, 1% penicillin–streptomycin and 1% pluronic F-68 in PBS). Each sample was layered on top of a histopaque layer at room temperature and underwent centrifugation at 1500 RPM for 15 min with zero deceleration. The cloudy interphase was carefully extracted, washed with FACS buffer, and centrifuged. The cells were then stained with fluorochrome-conjugated antibodies against CD45, TER119, Tie2, αV integrin, Thy, 6C3, CD 105, and CD 200. We isolated BCSPs using FACS and collecting cells that were CD45−, Ter119–, Tie2–, αV+, Thy–, 6C3–, CD105+, CD 200–. Sorts were performed in “purity” mode (BD FACS Aria II). For an apoptosis marker, Annexin V:FITC apoptosis detection kit II (BD Biosciences, San Jose, CA) was used according to manufacturer’s instructions. 

### 2.4. Isolation of Human Bone Progenitors

The protocol was described previously [21]. Adult human femoral heads were obtained from six patients whose age ranged from 65 to 85 at Stanford University Hospital following guidelines of Institutional Review Board (IRB)-35711. Following surgical excision, the specimens were processed as described in the “Isolation of skeletal progenitors”. The cells were stained with fluorochrome-conjugated antibodies against CD45, CD235, CD31, CD202b, CD146, Podoplanin, CD164, and CD73 prior to analysis by fluorescence-activated cell sorting (FACS). Human BCSPs were isolated by selecting for CD45–CD235–CD31–CD202b–PDPN+CD146+CD73+CD164+ cells. Sorts were performed in “purity” mode (BD FACS Aria II).

### 2.5. Transcriptional Expression Assay

Microarray analysis was performed on progenitor cells which were isolated and sorted as described previously [20]. Briefly, these cells were sorted into Trizol (Life Technologies, Carlsbad, CA, USA) and processed to collect RNA according to manufacturer’s instruction with a RNeasy Micro Kit (Qiagen, Germany). RNA was amplified twice with RiboAmp RNA amplification kit (Thermo Fisher Scientific, Waltham, MA, USA), streptavidin-labeled, fragmented, and hybridized to Affymetrix 430–2.0 arrays (Affymetrix, Santa Clara, CA, USA). The arrays were scanned with the Genechip Scanner 3000 (Affymetrix, Santa Clara, CA, USA), the data were exported to DCHIP software and submitted to Gene Expression Commons (https://gexc.stanford.edu) (7 June 2012)where they were normalized against a large registry of gene expression data, as reported previously [22]. 

### 2.6. Mechanical Strength Testing

Femurs were harvested 4 weeks post-fracture. Mechanical Strength Testing was performed on a SSM-EEX-50 delaminator by the R.H. Dauskardt Laboratory at Stanford University. Samples were preloaded to 1 Newton (N) and underwent three-point bending at a compression rate of 2 micrometer/second. The maximal load to fracture was recorded as “load” as a measure of mechanical strength. A minimum of three or more mice were used as the sample size of this study.

### 2.7. Histological Analysis

Tibias were harvested and fixed in 4% paraformaldehyde at 4 °C for 12 h, followed by decalcification in 19% EDTA for 3–4 weeks, embedded in paraffin and cut in 10 μm thick sections. For histomorphometry, sections were stained with Safranin O and Aniline blue to highlight cartilage and bone, respectively, and then imaged. One image out of every eight was used to quantify cartilage and bone formation. For TUNEL staining, tissue sections were treated with Ficin (Invitrogen, Carlsbad, CA, USA) for 7 min at 37 °C, endogenous peroxidase activity was blocked with 3% hydrogen peroxide in PBS for 10 min at room temperature. TUNEL staining was performed using the in situ Cell Death Detection Kit, POD (Roche, Indianapolis, IN, USA) according to manufacturer’s instructions. 

### 2.8. Radiographic Analysis

Tibias were explanted from mice 4 weeks following fracture. For X-ray analysis, specimens were imaged using the Spectral Ami X Optical Imaging Platform with a 40 kV variable energy X-ray source (Spectral Instruments Imaging, LLC, Tucson, AZ, USA). For microCT analysis, the specimen was imaged using micro-computed tomography (eXplore CT 120 model).

### 2.9. Skeletal Cell Culture

FACS-isolated progenitor cells were harvested as previously described. Cells were maintained in αMEM with 10% FBS, 1% Penicillin Streptomycin in a 37 °C incubator at 2% O_2_ and 7.5% CO_2_. For colony formation units, 500 BCSP cells were plated in triplicate and cultured for 2 weeks. For the proliferation assay, one thousand BCSPs were plated in a multi-well plate and counted by microscope at 5, 10, 15, and 20 days after seeding. 

### 2.10. Generation of Native and Truncated Del1 Proteins

Using Gibson assembly, both native and RAD (truncation) alleles of Del1 *(Edil3)* were cloned into the pAc-gP67B baculovirus vector. Each construct encodes an N-terminal 6xHis tag. Sf9 insect cell cultures were prepared by seeding 12-well culture plates at 1.0 × 10^6^ cells/mL and incubating for 30 min. Co-transfection complexes were prepared with 100 ng linearized baculovirus DNA, 1 μg of *Edil3* construct, 50μL serum free SF900 III and 5 μL Cellfectin II (ThermoFisher Scientific, Waltham, MA, USA). Transfection complexes were added to cells and cultures were incubated for 4 h at 27 °C. After removal of transfection mix, 2 mL of SF900-III medium with FBS was added to each well and cultures were incubated for 7 days at 27 °C. P0 virus was harvested by collecting the supernatant after centrifugation; stocks were stored at 4 °C. Amplification of P0 virus stock was performed by inoculating mid-log phase (e.g., 2 × 10^6^ cells/mL) cultures of Sf9 with 0.5 mL of P0 virus. When cell viability reached 30%, P1 supernatant stocks were collected by centrifugation and stored. For protein expression, Sf9 cells were grown in suspension culture to a density of 2 × 10^6^ cells/mL and, following addition of amplified P1 virus stock (1 mL per 50 mL of cell suspension), cultures were incubated at 27 °C, with shaking at 90 RPM, for 72 h. Conditioned medium supernatants were harvested following centrifugation at 1000× *g* for 5 min. Del1 protein was purified to homogeneity using immobilized metal ion chromatography (HisTrap-HP; GE Life Sciences, Piscataway, NJ, USA) and size-exclusion chromatography (Superdex 200 10/30; GE Life Sciences).

### 2.11. Western blotting

Western analysis was performed by blotting SDS-PAGE gels (Criterion TGX stain-Free; Bio-Rad, Hercules, CA, USA) to Immobilon-P (Millipore, Burlington, MA, USA) and probing with rabbit anti-Edil3 antibody (Abcam, Cambridge, UK) for 3 h at room temperature at a dilution of 1:400. An anti-rabbit IgG-HRP conjugate (Promega, Madison, WI, USA), in combination with Super Signal West Pico reagent (Pierce) (Thermo Fisher Scientific, Waltham, MA, USA), was used for detection.

### 2.12. Statistical Analysis

Data analysis was performed using Student’s *t*-test with a two-tailed distribution. Statistical analyses were performed using GraphPad Prism 9 (GraphPad, California, CA, USA). Deviation was graphically displayed as the standard error of the mean. Statistical significance was assigned for *p* < 0.05. All data are expressed ± SD. All experiments were performed in triplicate in at least two independent experiments. All data points represent biological replicates. 

## 3. Results

### 3.1. Del1 Knockout Mice Exhibited Impaired Skeletal Regeneration

Because our previous data showed Del1 KO mice expressed their phenotype in response to injury, we examined its expression in long bone fracture (Figure 1B). The transgenic mice were previously described where the LacZ–Neomycin cassette was inserted in exon 1 of the Del1 gene to disrupt the expression of the Del1 protein. Considering the LacZ expression was driven by the Del1 promoter, we asked whether Del1 was expressed during fracture healing by staining for LacZ expression. Tibial fracture was performed on WT and KO mice and the positive X-gal staining of LacZ suggested that Del1 transcription was activated during the fracture healing process (Figure 1B). 

KO and WT mice underwent tibial fracture and were examined at 4 weeks post-fracture. Plain X-rays showed a smaller fracture callus in the KO mice (Figure 1C) and microCT were used to quantify the size differences of the callus (Figure 1D). Because we previously demonstrated that increased apoptosis in the articular cartilage was correlated with increased severity of osteoarthritis in a model of traumatic osteoarthritis, we asked whether we could also detect increased apoptosis within the hypertrophic chondrocytes of the fracture callus. We performed TUNEL staining at 2-weeks post-fracture when hypertrophic chondrocytes were most prevalent and showed a significant increase in the number of apoptotic cells in the fracture callus of KO mice compared to WT (Figure 1E). Histomorphometry analysis was performed on fracture calluses of KO and WT mice using Safranin O for cartilage and Aniline blue for bone (Figure 1F). At 2 weeks and 4 weeks post-fracture, KO mice showed significantly less formed cartilage and less formed bone, respectively. 

Since previous publications have identified modulation of immune cells as an important feature of Del1 in fracture healing, we examined the bone immunohistology for changes in immune cells: monocytes (F4/80), lymphocytes (CD45R) and neutrophils (Ly-6B.2), however we saw no clear differences in the numbers of immune cells (Appendix A). Additionally, there was no evidence of differences in vascularity of the fracture callus as examined by staining for endothelial cells (CD31). These studies suggest that the observed decline in KO fracture healing may not be due to differences in immune or vasculogenic cell localization. 

### 3.2. Del1 Is Necessary for Expansion of BCSPs (Bone Cartilage Stromal Progenitors)

The findings of a smaller fracture callus with increased apoptosis were strongly reminiscent of the findings seen in hind-limb irradiation as a model of diminished fracture healing [21,23] We hypothesized the fracture healing phenotype in the KO mice was mediated by an effect of Del1 on BCSPs. The lineage hierarchy of the BCSPs showed that BCSPs were differentiated from Skeletal Stem Cells (SSCs) and could also differentiate into PCP (pro-chondrocyte progenitors), bone and stromal cells. BCSPs, as a transit amplifying progenitor cells, are activated by injury and rapidly proliferate to generate new cartilage, bone and stromal cells [4]. Del1 expression in multiple skeletal cells, including BCSPs, was consistent in mice that aged from postnatal (3 days old), to adults (2 months old), and aged (2 years old) mice (Figure 2B), consistent with its essential role in BCSPs across all developmental ages. 

We examined BCSPs during fracture healing in KO and WT mice as this is the skeletal progenitor sub-population most involved in fracture healing. The number of BCSPs present in non-fractured femurs were equivalent in WT and KO mice (Figure 2C). However, once fractured, SSCs and BCSPs were activated and underwent proliferation. The normal expansion of SSCs was not affected in KO mice. However, the BCSP expansion was attenuated in the KO compared with WT mice (Figure 2C). 

Considering our previous findings that Del1 protects cells against apoptosis, we examined the apoptosis of BCSPS in the KO mice fracture callus. We used annexin V to mark apoptotic BCSPs and quantify them using FACS analysis. We found that the number of apoptotic BCSPs were significantly higher in KO mice (Figure 2D). When fracture occurs, there was a phenotypic change from unfractured BCSPs to fracture (f-BCSPs) as defined by the added expression of the cell surface marker CD49. The f-BCSPs display increased resistance to apoptosis and increased osteogenic potential. We examined whether there was any impact on this transition in KO mice compared to WT. We did not find any difference in activation of BCSPs (Figure 2D). 

We then cultured BCSPs collected from the fracture callus of KO and WT mice 7 days post fracture to assess in vitro colonization, osteogenic differentiation (Alizarin Red) and proliferation assay (Figure 2E). One thousand BCSPs were extracted from the fracture callus at 7 days post-fracture, sorted through FACS, then the sorted cells were placed in tissue culture plates for colonization, the osteogenic assay, and proliferation assay. After one week of cell culture incubation, colonies were quantified. The results showed that BCSPs from KO formed significantly fewer colonies than BCSPs from WT (Figure 2E(i)). We also assessed the ability of BCSPs to differentiate to osteogenic fates by staining with Alizarin Red. The results showed the BCSPs from the KO had less Alizarin Red staining than the BCSPs from WT. Finally, twenty BCSPs were placed in tissue culture plates, incubated for 20 days and the number of cells quantified. The results showed that the proliferation of BCSP in vitro is significantly higher in WT than KO. We conclude loss of Del1 led to less bone formation at the fracture callus due to decreased colonization, decreased osteogenic potential and reduced expansion of BCSPs. 

To elucidate the regulatory mechanisms underlying Del1 regulation of BCSPs, we constructed a gene regulatory network using microarray data and computational modeling. We identified Prg4 and Brwd1 to be key regulators of the network, as they directly or indirectly regulated the expression of target genes (Figure 2F). The resulting microarray data was analyzed using the previously described Gene Expression Commons database, allowing us to compare our results to existing microarray data. Dozens of genes were identified to have differential expression in WT and KO BCSPs, with 104 genes with decreased expression in KO, and 209 genes with increased expression. Among genes of known functions, stem cell markers and genes promoting cell proliferation were downregulated in KO BCSPs. Genes upregulated in KO BCSPs were involved in apoptosis and extracellular matrix (ECM) formation (Figure 2G). Cumulatively, these data showed that the lack of Del1 protein changed the fate of BCSPs from proliferation to terminal differentiation and forming ECM. 

### 3.3. Del1 Activation of BCSP Expansion Is Dependent on the FAK Modulating RGD Domain

Del1 is a modular protein with multiple potential signaling motifs. We hypothesized that integrin signaling was primarily responsible for mediating BCSP proliferation. The binding domain of the native Del1 was engineered to RAD from native RGD with a baculovirus system (Figure 3A). Truncated Del1 from RGD to RAD displays an additional band with both Coomassie staining and Western blot (Figure 3B). When we placed the native Del1 protein or RAD mutant protein on hydrogels at the fracture callus, we found only native Del1 protein had an effect on increasing in vitro proliferation of BCSPs from KO (Figure 3C(i)) and in vivo (Figure 3C(ii)). 

The focal adhesion kinase (FAK) pathway plays a role in the regulation of multiple cellular processes that involve integrating extracellular signals from the extracellular matrix and growth factors. There are studies suggesting the FAK pathway is activated upon binding of Del1 to integrins. Therefore, we tested the effects of the FAK inhibitor (FAKi) on Del1-promoted BCSP proliferation (Figure 3D). Our in vitro cell proliferation results showed that FAKi would block the Del1 effects on BCSP from KO mice. Therefore, we concluded that integrin signaling through FAK is required for Del1′s ability to promote BCSP expansion in the fracture callus. 

### 3.4. Exogenous Del1 Rescues Proliferation of BCSP from KO Mice and Aged Human Skeletal Tissue

We tested whether the addition of exogenous Del1 to KO BCSPs would restore them to WT levels of activity in mice. We cultured isolated BCSPs from KO and WT mice at 7 days post-fracture following femur fracture. KO BCSPs cultured on Del1-coated plates were compared to KO and WT BCSPs cultured on bovine serum albumin (BSA)-coated plates as a control. KO BCSPs cultured on BSA had significantly fewer cell numbers than WT, but the addition of Del1 allowed KO BCSPs to expand at the same rate as WT BCSPs (Figure 4B). We also tested the ability of exogenous Del1 on promoting KO mice fracture healing in vivo. The highly crosslinked poly(ethylene glycol) hydrogel containing Del1 was able to slowly release protein at constant levels over a period of at least 4 weeks (Appendix A). We placed the Del1-releasing hydrogel adjacent to the femur fracture site at the time of surgery in KO mice. As controls, blank hydrogels were placed at the fracture site of Del1 KO and WT mice. Mice were harvested 4 weeks after surgery and analyzed for callus size with microCT. MicroCT analysis showed that the addition of Del1 protein led to significantly increased bone in the KO fracture callus compared to the control (Figure 4C). However, our results also showed that the rescue effects of Del1 on KO fracture could not be fully restored to the level of the WT. We also tested whether addition of Del1 led to increased bone strength in the healed bones. Using three-point bending to test the maximal load, we found the results were similar to the microCT analysis. Del1-treated KO fracture femurs were significantly stronger than the control KO (Figure 4C(ii)). Both results suggest the local delivery of Del1 led to only partial restoration of fracture healing. We hypothesized this was due to the low dosage delivery of Del1 protein as the exogenously applied protein may be less capable of penetrating through the matrix of the fracture callus compared to locally synthesized and released Del1. 

### 3.5. Increased Proliferation of Aged Human BCSPs with Del1

The data presented above showed the profound effects of Del1 on BCSPs from mice. To test whether Del1 has the same effects on human BCSPs, we treated isolated human BCSPs with human Del1 protein. Femur heads were obtained from aged patients (age from 65 to 85 years old) undergoing hip replacement surgery due to osteoarthritic disease. The human BCSPs were collected and sorted as described in Methods. The sorted human BCSPs were then cultured on plates coated with recombinant human Del1 or BSA as a control. The in vitro proliferation assay was performed by seeding sorted human BCSPs into a tissue culture plate and cell numbers were counted daily. Del1 mediated increase in human BCSPs proliferation became statistically significant on day 5 and continued to be significant on day 6 and 7 (Figure 4E). These results were reproduced in human BCSPs from multiple patients. 

## 4. Discussion

In our previous work, we demonstrated that Del1 has a crucial role in skeletal biology by protecting chondrocytes from apoptosis in a model of osteoarthritis [18]. Building on this finding and considering Del1′s expression during endochondral ossification, we hypothesized that Del1 may have a broader function in fracture healing. To investigate this, we utilized an internally fixed mouse femur fracture model and observed diminished bone formation and a smaller callus in Del1 knockout (KO) mice compared to wild-type (WT) mice. This led us to propose that Del1 might have an effect on BCSPs, given the newly discovered role of BCSPs in amplifying cell numbers for efficient callus formation. Our results revealed that there were significantly fewer BCSPs and increased apoptosis in the Del1 KO fracture calluses leading to impaired callus expansion, suggesting a critical role of Del1 in maintaining BCSPs. Furthermore, this defect was partially restored by in vivo local delivery of Del1. In vitro experiments also demonstrated that Del1 is necessary for BCSP expansion and could protect BCSPs from apoptosis. Interestingly, we also found that human Del1 could promote proliferation of human BCSPs, further emphasizing the relevance of Del1 in human skeletal biology.

These findings were further supported by our mRNA expression analysis of BCSPs harvested from the fracture site, which revealed that the absence of Del1 led to downregulation of cell proliferation-related genes and upregulation of pro-apoptosis related and ECM-related genes, which may be attributed to the role of BCSPs in the fracture healing process. Truncated Del1 constructs indicated that Del1 signals to BCSPs through the RGD domain to activate the FAK pathway. Taken together, our findings demonstrate that Del1 serves as a crucial proliferative factor for BCSPs in skeletal injury, particularly in their role as transit amplifying progenitor cells. 

The role of Del1 in skeletal biology has been a subject of investigation for years, although its precise functions are not yet fully understood. Previous studies have focused on Del1′s modulation of immune cells and its regulatory effects on immune plasticity in various inflammatory disorders such as periodontitis, sclerosis, and pulmonary disorders [11,19,24,25]. These studies have shown that Del1 plays a role in promoting the proliferation of hematopoietic stem/progenitor cells, regulating cell niche, and influencing cell fate determination. Notably, Del1 has been reported to restrain osteoclastogenesis and prevent bone loss in a primate model of periodontitis [26]. A separate study has shown a similar function of Del1 in periodontitis in mice. However, while these findings are related to skeletal tissue, the direct impact of Del1 on skeletal cells has been limited. One study using a cultured cell line and primary calvarial osteoblasts found that Del1 promotes osteoblast differentiation in vitro [16]. This effect was mediated through the RGD domain, which activated the FAK pathway which activates the Runx2 pathway to promote osteogenesis [27], consistent with our study’s findings. Our in vivo investigations of long bone fractures extend these previous findings and suggest that Del1 has a direct effect on resident progenitor cells that are required for bone regeneration by promoting their proliferation. However, it is important to note that our understanding of the molecular mechanisms underlying Del1 regulation in skeletal tissue remains limited at this stage. Further investigation is required to explore potential involvement of additional pathways beyond Runx2, such as Wnt, as reported in neurons, to gain a comprehensive understanding of Del1′s regulatory effects [28]. 

Although previous studies have suggested that the absence of Del1 would significantly impact inflammatory responses [9,14,24,26,29,30,31,32], our experimental model did not reveal significant differences in osteoclast numbers or localization as determined by TRAP staining. Similarly, immunohistochemical staining of multiple immune cell types did not exhibit notable differences in cell localization or numbers. However, these findings do not exclude the possibility that the absence of Del1 may subtly affect long bone fracture healing by regulating immune cells. Regardless of whether Del1 influences immune response during fracture healing, our findings strongly support the notion that Del1 promotes the expansion of BCSP and is at least in part responsible for the phenotype. 

In conclusion, the identification of Del1 as a key regulator of BCSP expansion and apoptosis protection highlights its crucial role in ensuring sufficient bone formation during fracture healing. Expanding our knowledge of Del1′s impact on aged human BCSPs opens up promising avenues for therapeutic interventions, offering significant potential for improving bone healing outcome and addressing clinical challenges associated with impaired bone regeneration. 

## Figures and Tables

**Figure 1 biomolecules-13-01214-f001:**
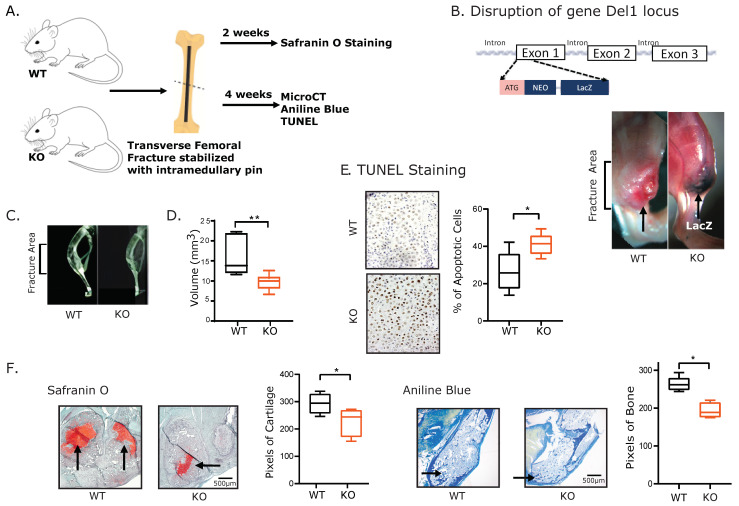
Del1 knockout mice exhibited impaired skeletal regeneration. (**A**) Schematic of the fracture creation and following assessment by plain X-rays, microCT, immunochemical analysis (Safranin O and Aniline Blue), and TUNEL staining. (**B**) Gene construct of Del1 knockout (KO) by inactivating the first ATG in the exon1 through insertion of the LacZ-Neomycin cassette at the PstI site (Top). The fractured tibia was stained 4 weeks post-fracture with whole mount with Xgal staining on LacZ (Bottom). The dark blue staining from KO mice(right) and the absence of the staining from the WT control (left) illustrated successful insertion of LacZ-Neomycin cassette which represented the successful disruption of Del1 expression. (**C**) Plain X-rays of the callus formation at 4 weeks post tibial fracture in WT and KO mice at 4 weeks post-fracture. (**D**) MicroCT scan of the callus formation at 4 weeks post tibial fracture to access the volume of fracture callus expansion in KO and WT mice. (**E**) TUNEL staining were used to quantify the apoptotic cells of the fracture callus at 2 weeks post-fracture in WT and KO (41.11 ± 2.30%versus 26.66 ± 4.17%). (**F**) Micrographs of immunochemical staining by Safranin O (Left) and Aniline Blue (Right) was used to visualize and quantify the cartilage and bone formation in KO and WT mice. Data are expressed as means ± SD. Significance of the differences statistically * *p* < 0.05, ** *p* < 0.01.

**Figure 2 biomolecules-13-01214-f002:**
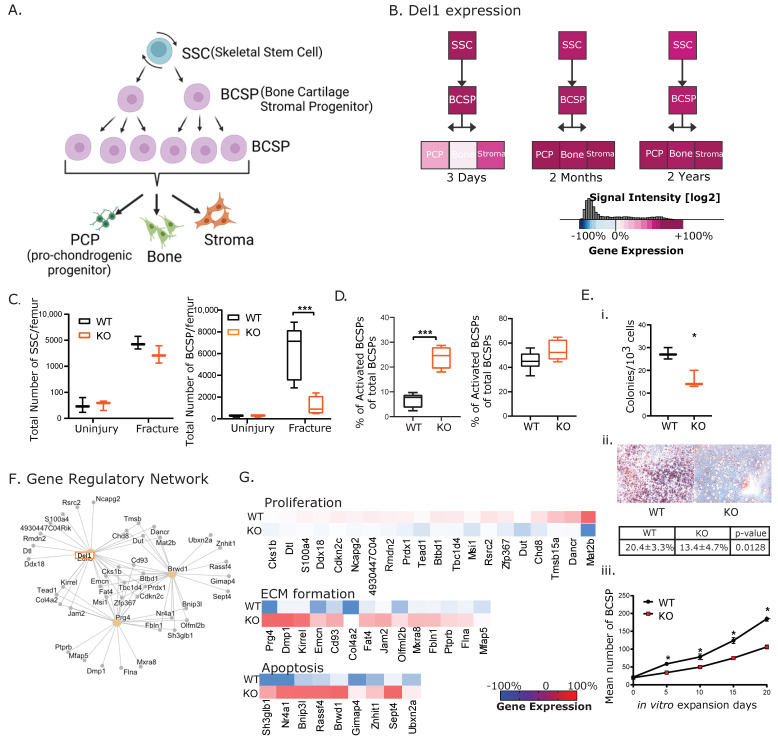
Del1 is necessary for expansion of BCSP (Bone Cartilage Stromal Progenitor). (**A**) Schematic of BCSP (Bone Cartilage Stromal Progenitor) lineage hierarchy: SSC (skeletal stem cell marked by CD45−TER119−TIE2−ITGAV+THY1+6C3−CD105−CD200+), BCSP (Bone, Cartilage, Stromal Progenitor marked by CD45−TER119−TIE2−ITGAV+THY1+6C3−CD105+, PCP (pro-chondrogenic cell marked by CD45+TER119−TIE2−ITGAV+THY1+6C3−CD105+CD200+), Bone (marked by CD45−TER119−TIE2−ITGAV+THY1+6C3−CD105+CD200−), and stroma cells (marked by CD45−TER119−TIE2−ITGAV+THY1−6C3−CD105+CD200−). (**B**) Cells were extracted, sorted and analyzed by microarray according to the methods. Del1 expression level was analyzed from mice of postnatal (3 days old), adult (8 weeks old), and aged (2 years old). The result suggested the expression of Del1 in mutliple skeletal cells were constant throughout the lifespan of mice. (**C**) Absolute numbers of SSC (Left) and BCSP (Right) from fractured femur was quantified by flow cytometry. The differences illustrated SSC from both WT and KO underwent expansion in similar fashion while expansion of BCSP were significantly compromised in KO compared with WT. (**D**) The percent of apoptotic BCSP (Anx V+) and activated BCSP(CD49+) out of total BCSPs was quantified through flow cytometry. It illustrated that KO did not impact the activation of BCSP but caused up-regulation of apoptotic activity of BCSP. (**E**) BCSP of KO and WT callus were sorted as described in Methods and seeded in vitro to access colony formation capability (i), osteogenic differentiation by Alizarin Red (ii), and in vitro proliferation (iii). (**F**) Gene regulatory network illustrated the interactions between Del1 and transcription factors. (**G**) Heatmap showing the expression levels of genes involved in extracellular matrix (ECM) formation, including prg4 and dimp1; and in apoptosis, including sh3glb1 and nr4a1; and in proliferation, including mat2b, dancr, and tmsb15a. Red indicates up regulation and blue indicates down regulation. Significance of the differences statistically * *p* < 0.05, *** *p* < 0.001.

**Figure 3 biomolecules-13-01214-f003:**
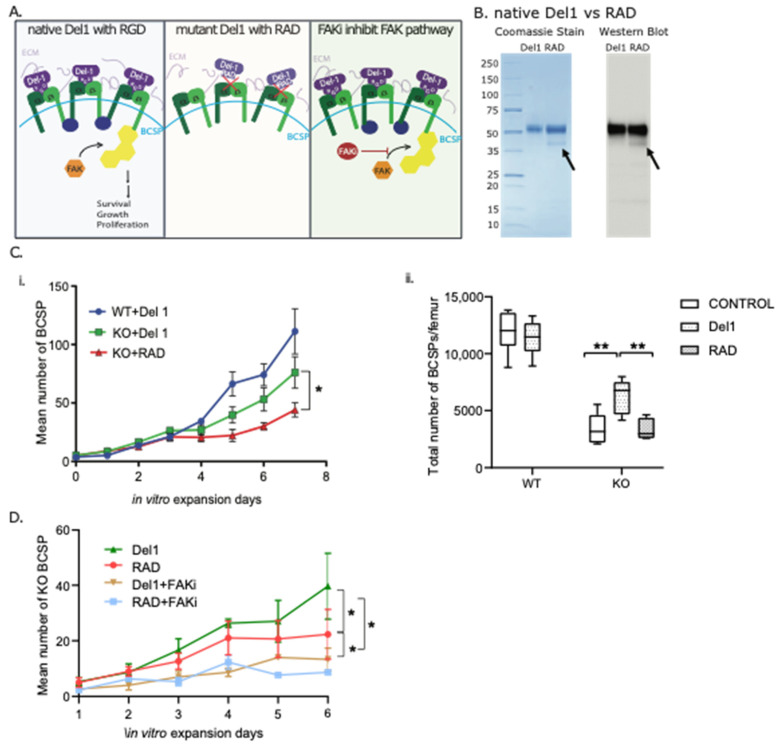
Del1 activation of BCSP expansion is dependent on the FAK modulation RGD domain. (**A**) Schematic demonstration of native Del1 with RGD motif (Del1) activates FAK pathway (left), truncated Del1 mutant with RAD motif (RAD) failed to bind to the integrin (middle), and FAKi inhibit FAK pathway activated by native Del1. (**B**) Coomassie stain (left) and western blot (right) following purification of native Del1 with RGD (Del1) and mutant Del1 with RAD motif (RAD) confirmed that native Del1 was successfully truncated to become mutant RAD-Del1 highlighted by the extra band pointed by arrow. (**C**) (i) BCSP extracted from KO or WT mice were seeded on Del1 or RAD treated cell surface. The proliferation of BCSP was counted for 7 days (Left). (ii) Del1, RAD or PBS of hydrogel were used to treat femur fracture in WT and KO mice at the time and site of fracture. The total number of BCSP in the fracture callus was quantified by FACS analysis at 7 days post-fracture. (**D**) BCSP from KO mice were extracted, sorted, and seeded in vitro on Del1 or RAD-coated surface, and then treated with FAKi or PBS. The results showed that Del1 upregulated the proliferation of KO BCSP significantly more than RAD both of which could be inhibited by FAKi. Significance of the differences statistically * *p* < 0.05, ** *p* < 0.01.

**Figure 4 biomolecules-13-01214-f004:**
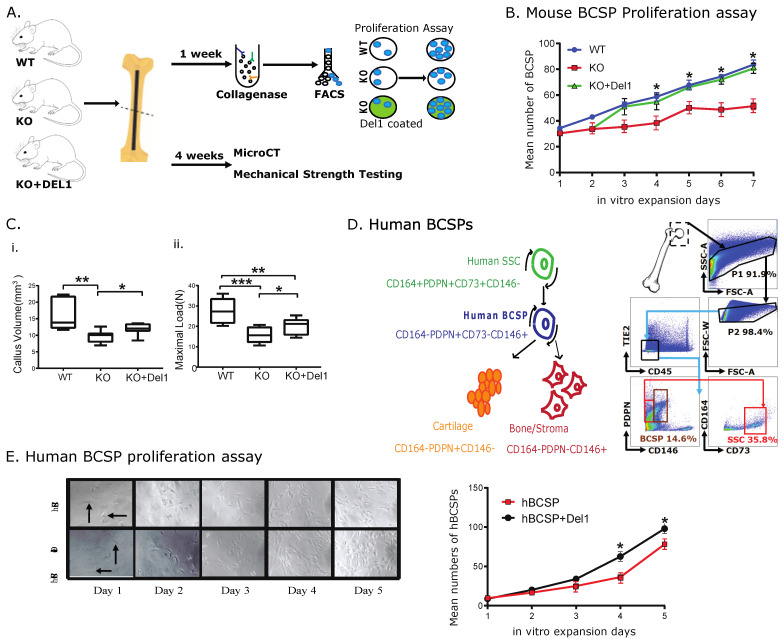
Exogenous Del1 rescues proliferation of BCSP from KO mice and aged human skeletal tissue. (**A**) Schematic of fracture creation and assessment of exogenous Del1 rescuing KO femur fracture impairment in vivo and in vitro. (**B**) BCSPs from WT and KO mice were sorted according to the Methods and seeded on PBS- or Del1- coated cell culture plate. In vitro proliferation assay was used to access the proliferation impacts of Del1 on KO BCSPs. (**C**) Exogenous Del1 in hydrogel was implanted at the time and the position of fracture creation on femur in vivo. The healing outcome was evaluated 4 weeks post-fracture by microCT (i) and mechanical strength testing (ii). Callus volume for each group: WT: 16.3 ± 5.1 mm^3^ vs. KO 9.64 ± 1.94 mm^3^ vs KO+ Del1 11.88 ± 1.77 mm^3^, n = 5. The mechanical strength testing showed WT 26.84 ± 5.9 N vs. KO 16.06 ± 3.8 N vs. KO+Del1 20.1 ± 4.02 N, n = 5. (**D**) It illustrated the surface markers and gating strategy to identify and sort for human BCSPs. Femoral head were obtained from patients whose ages range from 65 to 85 and processed described in the “Isolation of human bone progenitors” section of Methods. (**E**) Human BCSPs were sorted and then seeded on PBS- or Del1 coated surface followed by an assay of proliferation. The image of seeded cells were shown on left and the number of proliferated cells were quantified on right. Data are expressed as means±SD. Significance of the differences statistically * *p* < 0.05, ** *p* < 0.01, *** *p* < 0.001.

## Data Availability

All data that support the findings of this study are available from the corresponding author upon reasonable request.

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
