# Peer review of "Del1 Is a Growth Factor for Skeletal Progenitor Cells in the Fracture Callus"

_biomolecules, 2023, doi:10.3390/biom13081214_

Round 1

Reviewer 1 Report

I dont have suggestions. The work is wonderfool 

Author Response

Thank you for taking the time to review our manuscript. 

Reviewer 2 Report

This is a very interesting manuscript where the authors have examined the role of Del1 as a growth factor for skeletal progenitor cells in the fracture callus. The report is well written with clear objectives and clearly presented results. 

There are a couple of things the authors need to address.

In the abstract "Del1 injections may improve the success of orthopedic surgeries and fracture healing by enhancing the proliferation and survival of BCSPs, which are crucial for generating new bone tissue during the process of bone..."  Could this depend on the timing of the injection as it appears that Del1 has a very precise influence on the cells in the development of the fracture repair. Could the authors comment further on this.

I may have missed it upon reading the manuscript but with the mice fracture models what were the n numbers of the groups of mice used? 

Line 240 typo should be (3days old) 

Fig4d There is a large black bar in the graph

Line 253 - grammar. "When fracture occurred..." 

Line267-8 - Clarify this statement "...were seeded on tissue culture,"

How was the Del1 incorporated into the hydrogel?

Fig 4e Cannot read the y-axis labels

English language is good. A couple of grammar corrections and a typo to correct.

Author Response

Thank you for taking the time to review our manuscript entitled: Del1 is a growth factor for skeletal progenitor cells in the fracture callus. I appreciate your thoughtful feedback, and I am grateful for the opportunity to respond to your comments.

I would like to address the questions you raised regarding the potential role of Del1 injections may improve the success of orthopedic surgeries and fracture healing.

Response: The timing of Del1 injection plays a crucial role in fracture healing due to the interconnectedness and mutual influence of the inflammatory responses and skeletal formation process. The optimal efficacy of Del1 in promoting healing may be restricted to a narrow timeframe. However, in our study, we administered Del1 to mice in vivo using a hydrogel delivery system, enabling a sustained release of Del1 throughout the healing process. It is worth noting that mouse fracture heals relatively quickly with rapid resolution of inflammation compared to humans. Even though we demonstrated Del1 promoted the proliferation of human skeletal progenitor cells, further experiments are necessary to determine the effectiveness and appropriate timing of Del1 administration in human subjects.

As for the n number of the groups of mice used, we used minimal 3 mice per group for the fracture models in flow cytometry and mechanical strength testing. I have added this information at the end of the ‘Fracture model’ and ‘Mechanical strength testing’ in Methods.

The typo in Line 240 has been corrected.

I do not find the black bar that is unnecessary. I assume you mean there is an arrow pointed from the femoral head to the FACS plotting. This arrow was used to demonstrate that the human samples were collected from patients’ femoral head and used for FACS sorting.

Line 256 grammar has been corrected.

Line 267-8 We meant to suggest the cells were sorted from FACS sorting and then plated on tissue culture plate.

As for the methods of incorporating Del1 into hydrogel, we dissolve Del1 protein with PBS and add the mixture into hydrogel solution.

The y-axis on the images is hBCSP and hBCSP+Del1, and the y-axis on graph is Mean numbers of BCSPs. The font size has been increased for improved readability.

Once again, I sincerely appreciate your time and effort in reviewing my paper. Your feedback has been invaluable in improving the quality and impact of my research. I hope that the revisions I have made address your concerns and that the updated version of the paper will meeting your expectation.

Reviewer 3 Report

(1) Is there an ethical approval number for animal experiments? If so, please provide it.

(2) Please explain why safranine O+Aniline blue staining is selected to display cartilage and bone. Generally, safranine O+fast green staining should be selected, and HE conventional staining should be used as the control.

(3) Please provide the dilution ratio of the antibody for Western blot experiments.

(4) The staining diagram of tissue slices should provide a scale.

(5) The discussion is too simplistic, and it is necessary to add appropriate references to explain the possible mechanisms behind the research results.

Author Response

We would like to express our sincere gratitude for your valuable feedback on our manuscript, titled: Del1 is a growth factor for skeletal progenitor cells in the fracture callus. We appreciate the time and effort you have dedicated to reviewing our work, and we are pleased to have the opportunity to address your concerns and incorporate your suggestions in our revised manuscript.

Firstly, we would like to address your comment regarding the ethical approval number. In the first section of the methods, we have included the information regarding our animal protocols proved by Stanford and UAB IACUC. Our protocol number is 15966 in Stanford and 21355 in UAB.

Secondly, we appreciate your suggestion regarding the immunohistology analysis of the cartilage and bone formation. The reason we choose Safranin alone instead of Safranin O/fast green for cartilage staining is to simplify the interpretation of results. By eliminating the fast green counterstain, the visual examination of stained sections becomes less complex, allowing for easier identification and quantification of proteoglycan content stained by Safranin O.

Regarding the dilution ratio of the western blot antibody, the dilution ratio of rabbit anti-Edil3 is 1:400. The methods for the western blot has been re-edited to include this information and more details.

The scale was added to the image.

Finally, we acknowledge your comment regarding the simplicity of our discussion. We agree with your observation that we do not discuss much about the regulatory pathways of Del1. That is because the current understanding of Del1’s molecular mechanisms and its impact on skeletal tissue is limited. However, we have made efforts to include relevant information regarding the pathways through which Del1 promotes osteogenesis. In the revised manuscript, we have added information regarding the activation of TGF-b pathway, specifically targeting Runx2, which is known to play a crucial role in osteogenesis. Our results also suggested the involvement of Del1 in activating the ab3 integrin, which in turn triggers the FAK pathway that upregulates Runx2 expression.

There are reports suggesting that Del1 can attenuate canonical Wnt signaling in neuron tissue. And Del1 has been found to be regulated by p53 in endothelial cells. However, it is important to note that the understanding of the molecular mechanisms regulating Del1 is still very limited. Therefore, we have decided not to delve extensively into the potential mechanisms beyond what we have shown through our results. We have added a brief content in the discussion section to provide further elaboration.

Once again, we thank you for your time and effort in reviewing our work. We believe that the revised manuscript now effectively addresses your concerns and contributes to the current understanding of Del1’s role in skeletal tissue.